

# Pre-trained convolutional neural networks as feature extractors toward improved malaria parasite detection in thin blood smear images

Sivaramakrishnan Rajaraman[1], Sameer K. Antani[1], Mahdieh Poostchi[1], Kamolrat Silamut[2], Md. A. Hossain[3], Richard J. Maude[2,4,5], Stefan Jaeger[1] and George R. Thoma[1]

[1] Lister Hill National Center for Biomedical Communications, National Library of Medicine, Bethesda, MD, United States of America
[2] Mahidol-Oxford Tropical Medicine Research Unit, Mahidol University, Bangkok, Thailand
[3] Department of Medicine, Chittagong Medical Hospital, Chittagong, Bangladesh
[4] Centre for Tropical Medicine and Global Health, Nuffield Department of Medicine, University of Oxford, Oxford, United Kingdom
[5] Harvard TH Chan School of Public Health, Harvard University, Boston, MA, United States of America

## ABSTRACT

Malaria is a blood disease caused by the *Plasmodium* parasites transmitted through the bite of female Anopheles mosquito. Microscopists commonly examine thick and thin blood smears to diagnose disease and compute parasitemia. However, their accuracy depends on smear quality and expertise in classifying and counting parasitized and uninfected cells. Such an examination could be arduous for large-scale diagnoses resulting in poor quality. State-of-the-art image-analysis based computer-aided diagnosis (CADx) methods using machine learning (ML) techniques, applied to microscopic images of the smears using hand-engineered features demand expertise in analyzing morphological, textural, and positional variations of the region of interest (ROI). In contrast, Convolutional Neural Networks (CNN), a class of deep learning (DL) models promise highly scalable and superior results with end-to-end feature extraction and classification. Automated malaria screening using DL techniques could, therefore, serve as an effective diagnostic aid. In this study, we evaluate the performance of pre-trained CNN based DL models as feature extractors toward classifying parasitized and uninfected cells to aid in improved disease screening. We experimentally determine the optimal model layers for feature extraction from the underlying data. Statistical validation of the results demonstrates the use of pre-trained CNNs as a promising tool for feature extraction for this purpose.

Corresponding author
Sivaramakrishnan Rajaraman, sivaramakrishnan.rajaraman@nih.gov

## INTRODUCTION

Malaria is a mosquito-borne blood disease caused by the *Plasmodium* parasites transmitted through the bite of the female Anopheles mosquito. Different kinds of parasites including *P. ovale, P. malariae, P. vivax and P. falciparum* infect humans; however, the effects of *P. falciparum* can be lethal. In 2016, the World Health Organization (WHO) reported 212 million instances of the disease across the world (*WHO, 2016*). Microscopic thick and thin blood smear examinations are the most reliable and commonly used method for disease diagnosis. Thick blood smears assist in detecting the presence of parasites while thin blood smears assist in identifying the species of the parasite causing the infection (*Centers for Disease Control and Prevention, 2012*). The diagnostic accuracy heavily depends on human expertise and can be adversely impacted by the inter-observer variability and the liability imposed by large-scale diagnoses in disease-endemic/resource-constrained regions (*Mitiku, Mengistu & Gelaw, 2003*). Alternative techniques such as polymerase chain reaction (PCR) and rapid diagnostic tests (RDT) are used; however, PCR analysis is limited in its performance (*Hommelsheim et al., 2014*) and RDTs are less cost-effective in disease-endemic regions (*Hawkes, Katsuva & Masumbuko, 2009*).

In the process of applying machine learning (ML) methods to medical data analysis, meaningful feature representation lies at the core of their success to accomplish desired results. A majority of image analysis-based computer-aided diagnosis (CADx) software use ML techniques with hand-engineered features for decision-making (*Ross et al., 2006*; *Das et al., 2013*; *Poostchi et al., 2018*). However, the process demands expertise in analyzing the variability in size, background, angle, and position of the region of interest (ROI) on the images. To overcome challenges of devising hand-engineered features that capture variations in the underlying data, Deep Learning (DL), also known as deep hierarchical learning, is used with significant success (*LeCun, Bengio & Hinton, 2015*). DL models use a cascade of layers of non-linear processing units to self-discover hierarchical feature representations in the raw data. Higher-level features are abstracted from lower-level features to aid in learning complex, non-linear decision-making functions, resulting in end-to-end feature extraction and classification (*Schmidhuber, 2015*). Unlike kernel-based algorithms like Support Vector Machines (SVMs), DL models exhibit improved performance with an increase in data size and computational resources, making them highly scalable (*Srivastava et al., 2014*).

For images, an important source of information lies in the spatial local correlation among the neighboring pixels/voxels. Convolutional Neural Networks (CNN), a class of DL models, are designed to exploit this information through the mechanisms of local receptive fields, shared weights and pooling (*Krizhevsky, Sutskever & Hinton, 2012*). In 2012, Alex Krizhevsky proposed AlexNet, a CNN based DL model that won the ImageNet Large Scale Visual Recognition Challenge (ILSVRC) and substantially boosted the performance of CNNs toward classifying natural images (*Krizhevsky, Sutskever & Hinton, 2012*). Several representative CNNs like VGGNet (*Simonyan & Zisserman, 2015*), GoogLeNet (*Szegedy et al., 2014*), and ResNet (*He et al., 2016*) demonstrated significant improvements in succeeding ILSVRC annual challenges. A model named Xception was proposed that uses

depth-wise separable convolutions (*Chollet, 2016*) to outperform the Inception-V3 model (*Szegedy et al., 2016*) on the ImageNet (*Deng et al., 2009*) data classification task. A CNN variant called Densely Connected Convolutional Networks (DenseNet) was proposed (*Huang et al., 2016*) that utilizes a network architecture in which each layer is directly connected to every later layer. The model has achieved noteworthy improvements over the state-of-the-art while using significantly fewer parameters and computations.

The promising performance of CNNs is accompanied by the availability of a huge amount of annotated data. With scarcity for annotated medical imagery, Transfer Learning (TL) methods are used where pre-trained DL models are either fine-tuned on the underlying data or used as feature extractors to aid in visual recognition tasks (*Razavian et al., 2014*). These models transfer their knowledge gained while learning the generic features from large-scale datasets like ImageNet to the underlying task. The transfer of previously-learned skills to a new situation is generalized, rather than unique to the situation. Since the results published in *Razavian et al. (2014)*, it is recognized that CNNs trained on large-scale datasets could serve as feature extractors for a wide range of computer vision tasks to aid in improved performance, as compared to state-of-the-art methods (*Bousetouane & Morris, 2015*).

At present, researchers across the world have begun to apply DL tools and obtain promising results in a wide variety of medical image analyses/understanding tasks (*Rajaraman et al., 2017*; *Suzuki, 2017*). Literature also reveals studies pertaining to applying DL methods to the task of malaria parasite detection. *Dong et al. (2017)* compared the performance of SVM and pre-trained DL models including LeNet (*LeCun et al., 1998*), AlexNet, and GoogLeNet toward classifying parasitized and uninfected cells. The authors segmented the red blood cells (RBCs) from thin blood smear images and randomly split into train and test sets. A total of 25% of the training images were randomly selected to validate the models. *Liang et al. (2017)* proposed a 16-layer CNN toward classifying the uninfected and parasitized cells. Features were extracted using the pre-trained AlexNet and an SVM classifier was trained on the extracted features. The performance of the proposed model was compared to that of the pre-trained CNN. The study reported that the custom model was more accurate, sensitive and specific than the pre-trained model. Images were resampled to 44 × 44 pixel resolution to compensate for the lack of computational resources. *Bibin, Nair & Punitha (2017)* proposed a 6-layer deep belief network toward malaria parasite detection in peripheral blood smear images. The authors reported 96.4% accuracy in the task of classifying a dataset of 4,100 cells with randomized train/test splits. *Gopakumar et al. (2018)* employed a customized CNN model for analyzing videos containing a focus stack of the field of views of Leishman stained slide images toward the process of automated parasite detection. The authors used a customized portable slide scanner and off-the shelf components for data acquisition and demonstrated sensitivity and specificity of 97.06% and 98.50% respectively. In summary, existing DL studies have been evaluated on relatively small image sets and/or randomized train/test splits. None of the studies have reported the performance of the predictive models at the patient level. Although the reported outcomes are promising, existing approaches need to substantiate their robustness on a larger set of images with cross-validation studies at the patient level. Evaluation on patient-level
provides a more realistic performance evaluation of the predictive models as the images in the independent test set represent truly unseen images for the training process, with no information about staining variations or other artifacts leaking into the training data. This would help to reduce bias and generalization errors. Tests for statistically significant differences in performance would further assist in the process of optimal model selection prior to deployment. It is reasonable to mention that the state-of-the-art still leaves much room for progress in this regard.

In this work, we evaluated the performance of pre-trained CNN based DL models as feature extractors toward classifying the parasitized and uninfected cells to aid in improved disease screening. The important contributions of this work are as follows: (a) presentation of a comparative analysis of the performance of customized and pre-trained DL models as feature extractors toward classifying parasitized and uninfected cells, (b) cross-validating the performance of the predictive models at the patient level to reduce bias and generalization errors, (c) analysis and selection of the optimal layer in the pre-trained models to extract features from the underlying data, and (d) testing for the presence/absence of a statistically significant difference in the performance of customized and pre-trained CNN models under study. The following paper is organized as follows: 'Materials and Methods' elaborates on the materials and methods, 'Results' presents the results, and 'Discussions and Conclusion' discusses the results and concludes the paper.

## MATERIALS AND METHODS

### Data collection

To reduce the burden for microscopists in resource-constrained regions and improve diagnostic accuracy, researchers at the Lister Hill National Center for Biomedical Communications (LHNCBC), part of National Library of Medicine (NLM), have developed a mobile application that runs on a standard Android® smartphone attached to a conventional light microscope (*Poostchi et al., 2018*). Giemsa-stained thin blood smear slides from 150 *P. falciparum*-infected and 50 healthy patients were collected and photographed at Chittagong Medical College Hospital, Bangladesh. The smartphone's built-in camera acquired images of slides for each microscopic field of view. The images were manually annotated by an expert slide reader at the Mahidol-Oxford Tropical Medicine Research Unit in Bangkok, Thailand. The de-identified images and annotations are archived at NLM (IRB#12972). We applied a level-set based algorithm to detect and segment the red blood cells (*Ersoy et al., 2012*).

### Cross-validation studies

The dataset consists of 27,558 cell images with equal instances of parasitized and uninfected cells. Positive samples contained *Plasmodium* and negative samples contained no *Plasmodium* but other types of objects including staining artifacts/impurities. We evaluated the predictive models through five-fold cross-validation. Cross-validation has been performed at the patient level to ensure alleviating model biasing and generalization errors. The count of cells for the different folds is shown in Table 1.

**Table 1 Data for cross-validation studies.**

| Folds | Parasitized | Uninfected |
|---|---|---|
| 1 | 2,756 | 2,757 |
| 2 | 2,758 | 2,758 |
| 3 | 2,776 | 2,762 |
| 4 | 2,832 | 2,760 |
| 5 | 2,657 | 2,742 |
| **Total** | **13,779** | **13,779** |

The images were re-sampled to $100 \times 100$, $224 \times 224$, $227 \times 227$ and $299 \times 299$ pixel resolutions to suit the input requirements of customized and pre-trained CNNs and normalized to assist in faster convergence. The models were trained and tested on a Windows® system with Intel® Xeon® CPU E5-2640v3 2.60-GHz processor, 1 TB HDD, 16 GB RAM, a CUDA-enabled Nvidia® GTX 1080 Ti 11GB graphical processing unit (GPU), Matlab® R2017b, Python® 3.6.3, Keras® 2.1.1 with Tensorflow® 1.4.0 backend, and CUDA 8.0/cuDNN 5.1 dependencies for GPU acceleration.

## Customized model configuration

We also evaluated the performance of a customized, sequential CNN in the task of classifying parasitized and uninfected cells toward disease screening. We propose a sequential CNN as shown in Fig. 1, similar to the architecture that *LeCun & Bengio (1995)* advocated for image classification.

The proposed CNN has three convolutional layers and two fully connected layers. The input to the model constitutes segmented cells of $100 \times 100 \times 3$ pixel resolution. The convolutional layers use $3 \times 3$ filters with 2 pixel strides. The first and second convolutional layers have 32 filters and the third convolutional layer has 64 filters. The sandwich design of convolutional/rectified linear units (ReLU) and proper weight initialization enhances the learning process (*Shang et al., 2016*). Max-pooling layers with a pooling window of $2 \times 2$ and 2 pixel strides follow the convolutional layers for summarizing the outputs of neighboring neuronal groups in the feature maps. The pooled output of the third convolutional layer is fed to the first fully-connected layer that has 64 neurons, and the second fully connected layer feeds into the Softmax classifier. Dropout regularization (*Srivastava et al., 2014*) with a dropout ratio of 0.5 is applied to outputs of the first fully connected layer. The model is trained by optimizing the multinomial logistic regression objective using stochastic gradient descent (SGD) (*LeCun, Bengio & Hinton, 2015*) and Nesterov's momentum (*Botev, Lever & Barber, 2017*). The customized model is optimized for hyper-parameters by a randomized grid search method (*Bergstra & Bengio, 2012*). We initialized search ranges to be [1e−7 5e−2], [0.8 0.99] and [1e−10 1e−2] for the learning rate, SGD and L2-regularization parameters, respectively. We evaluated the performance of the customized model in terms of accuracy, Area Under Curve (AUC), sensitivity, specificity, F1-score (*Lipton, Elkan & Naryanaswamy, 2014*) and Matthews correlation coefficient (MCC) (*Matthews, 1975*).

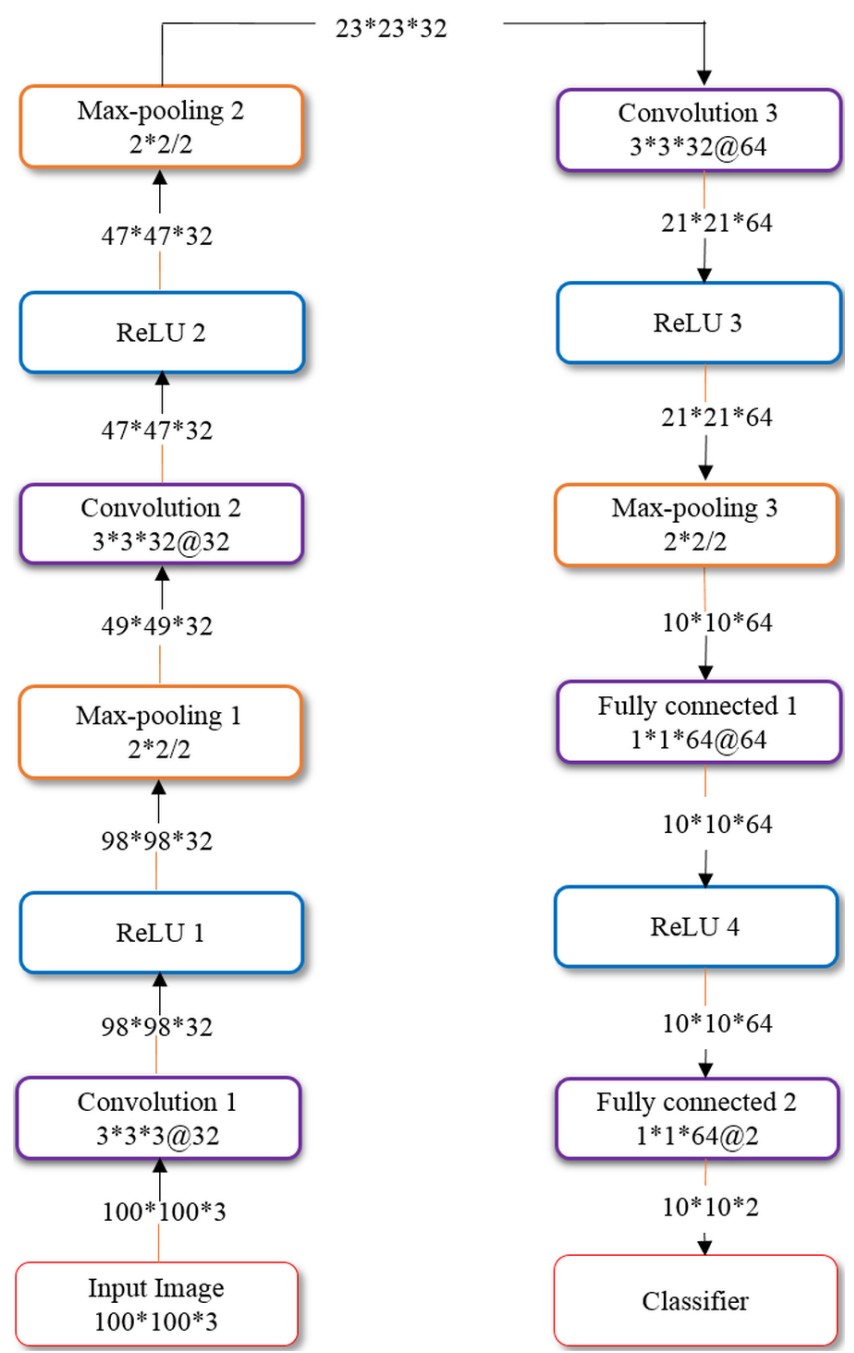

**Figure 1  Architecture of the customized model.**

## Feature extraction using pre-trained models

We evaluated the performance of pre-trained CNNs including AlexNet (winner of ILSVRC 2012), VGG-16 (winner of ILSVRC's localization task in 2014), Xception, ResNet-50 (winner of ILSVRC 2015) and DenseNet-121 (winner of the best paper award in CVPR 2017) toward extracting the features from the parasitized and uninfected cells. The models

were optimized for hyper-parameters by the randomized grid search method. We initialized search ranges to be [1e−5 5e−2], [0.8 0.99] and [1e−10 1e−2] for the learning rate, Nesterov's SGD and L2-regularization parameters, respectively. We instantiated the convolutional part of the pre-trained CNNs and trained a fully-connected model with dropout (dropout ratio of 0.5) on top of the extracted features. We also empirically determined the optimal layer for feature extraction to aid in improved classification. We evaluated the performance of the pre-trained CNNs in terms of accuracy, AUC, sensitivity, specificity, F1-score, and MCC. The model architecture and weights for the pre-trained CNNs were downloaded from GitHub repositories (*Chollet, 2017*; *Yu, 2016*).

## Statistical analysis

We performed statistical analyses to choose the best model for deployment. Statistical methods like one-way analysis of variance (ANOVA) are used to determine the presence or absence of a statistically significant difference between the means of three or more individual, unrelated groups (*Rossi, 1987*). One-way ANOVA tests the null hypothesis (H0) given by H0: $\mu_1 = \mu_2 = \cdots = \mu_k$ where $\mu$ = mean of parameters for the individual groups and $k$ = total number of groups. If a statistically significant result is returned by the test, H0 is rejected and the alternative hypothesis (H1) is accepted to infer that a statistically significant difference exists between the means of at least two groups under study. However, it would be appropriate to use this parametric test only when the underlying data satisfies the assumptions of independence of observations, absence of significant outliers, normality of data and homogeneity of variances (*Daya, 2003*). When the conditions are violated, a non-parametric alternative like Kruskal-Wallis H test (also called the one-way ANOVA on ranks) could be used (*Vargha & Delaney, 1998*). This is an omnibus test that couldn't identify the specific groups that demonstrate statistically significant differences in their mean values. A post-hoc analysis is needed to identify these groups that demonstrate statistically significant differences (*Kucuk et al., 2016*). We performed Shapiro–Wilk test (*Royston, 1992*) to check for data normality and Levene's statistic test (*Gastwirth, Gel & Miao, 2009*) to study the homogeneity of variances for the performance metrics for the different models under study. Statistical analyses were performed using IBM® SPSS® statistical package (IBM SPSS Statistics for Windows, Version 23.0; IBM Corp., Armonk, NY, USA).

# RESULTS

## Cell segmentation and detection

We applied a level-set based algorithm to detect and segment the RBCs as shown in Fig. 2. The first step is the cell detection where we applied a multi-scale Laplacian of Gaussian (LoG) filter to detect centroids of individual RBCs. The generated markers are used to segment the cells within a level set active contour framework to confine the evolving contour to the cell boundary. Morphology opening operation is applied as post-processing to remove false detected objects such as staining artifacts using average cell size. White blood cells (WBCs) are filtered out using a one-to-one correspondence based on cell ground-truth annotations since WBCs are not ROIs for this work. We have evaluated our

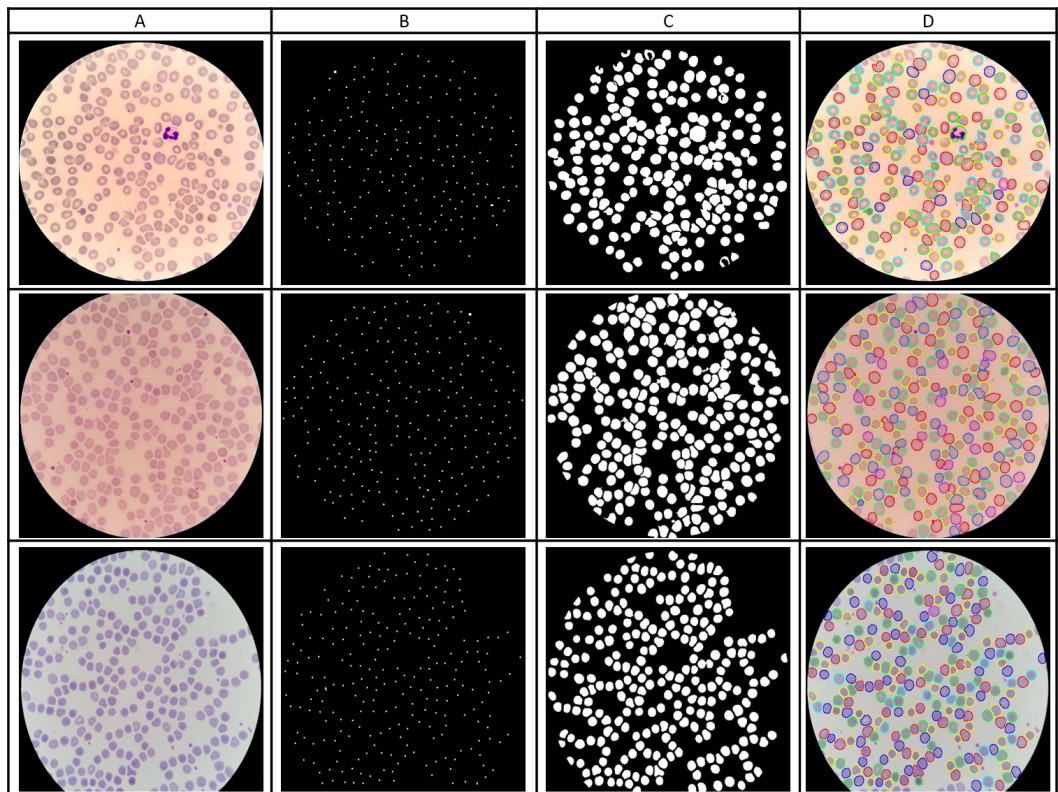

**Figure 2** **RBC detection and segmentation using level sets.** (A) Input image. (B) Initial cell detection using LoG. (C) Final RBC segmentation mask. (D) Segmentation results superimposed on the original image.

cell detection based on the manual point-wise annotation of infected and uninfected cells. To do so, we applied a one-to-one point matching scheme: For each segmented cell, we checked the number of manual ground-truth points in the segmented cell region. If there was exactly one point in the region, we counted this as a true positive (TP). If there was no point in the region, we counted this as a false positive (FP). If there was more than one point in the region, we considered this as an under-segmentation or false negative (FN). These counts then allowed us to compute the presented values for positive predictive value (PPV), sensitivity and F1-score. For cell detection, we obtained a PPV of 0.944, sensitivity of 0.962 and F1-score of 0.952.

## Performance metrics evaluation

For the customized and pre-trained models, we empirically determined the optimum value to be 0.9 and 1e–6 for the SGD momentum and L2-regularization, respectively. For the learning rate, we determined the optimum value to be 1e–5 and 1e–6 for the customized and pre-trained CNNs respectively. The second fully connected layer from AlexNet, VGG-16 and the last layer before the final classification layer from Xception, ResNet-50, and DenseNet-121 were selected for feature extraction. Table 2 lists the performance metrics achieved by the models in the process of classifying parasitized and uninfected

**Table 2  Performance metrics.**

| Models | Accuracy | AUC | Sensitivity | Specificity | F1-score | MCC |
|---|---|---|---|---|---|---|
| AlexNet | $0.937 \pm 0.012$ | $0.981 \pm 0.007$ | $0.940 \pm 0.017$ | $0.933 \pm 0.034$ | $0.937 \pm 0.011$ | $0.872 \pm 0.024$ |
| VGG-16 | $0.945 \pm 0.015$ | $0.981 \pm 0.007$ | $0.939 \pm 0.022$ | $0.951 \pm 0.019$ | $0.945 \pm 0.016$ | $0.887 \pm 0.030$ |
| ResNet-50 | $\mathbf{0.957 \pm 0.007}$ | $\mathbf{0.990 \pm 0.004}$ | $\mathbf{0.945 \pm 0.020}$ | $\mathbf{0.969 \pm 0.009}$ | $\mathbf{0.957 \pm 0.008}$ | $\mathbf{0.912 \pm 0.014}$ |
| Xception | $0.890 \pm 0.107$ | $0.948 \pm 0.062$ | $0.931 \pm 0.039$ | $0.835 \pm 0.218$ | $0.895 \pm 0.100$ | $0.772 \pm 0.233$ |
| DenseNet-121 | $0.931 \pm 0.018$ | $0.976 \pm 0.023$ | $0.942 \pm 0.023$ | $0.926 \pm 0.032$ | $0.931 \pm 0.017$ | $0.894 \pm 0.036$ |
| Customized | $0.940 \pm 0.010$ | $0.979 \pm 0.009$ | $0.931 \pm 0.026$ | $0.951 \pm 0.030$ | $0.941 \pm 0.010$ | $0.880 \pm 0.020$ |

**Notes.**
Bold text indicate the performance measures of the best-performing model/s.

**Table 3  Candidate layers giving the best performance.**

| Model | Optimal layer |
|---|---|
| AlexNet | fc6 |
| VGG-16 | block5_conv2 |
| ResNet-50 | res5c_branch2c |
| Xception | block14_sepconv1 |
| DenseNet-121 | Conv5_16_x2 |

**Table 4  Performance metrics achieved with feature extraction from optimal layers.**

| Models | Accuracy | AUC | Sensitivity | Specificity | F1-score | MCC |
|---|---|---|---|---|---|---|
| AlexNet | $0.944 \pm 0.010$ | $0.983 \pm 0.006$ | $0.947 \pm 0.016$ | $0.941 \pm 0.025$ | $0.944 \pm 0.010$ | $0.886 \pm 0.020$ |
| VGG-16 | $\mathbf{0.959 \pm 0.009}$ | $\mathbf{0.991 \pm 0.004}$ | $0.949 \pm 0.020$ | $0.969 \pm 0.016$ | $\mathbf{0.959 \pm 0.009}$ | $0.916 \pm 0.017$ |
| ResNet-50 | $\mathbf{0.959 \pm 0.008}$ | $\mathbf{0.991 \pm 0.005}$ | $0.947 \pm 0.015$ | $\mathbf{0.972 \pm 0.010}$ | $\mathbf{0.959 \pm 0.009}$ | $\mathbf{0.917 \pm 0.017}$ |
| Xception | $0.915 \pm 0.005$ | $0.965 \pm 0.019$ | $0.925 \pm 0.039$ | $0.907 \pm 0.120$ | $0.918 \pm 0.042$ | $0.836 \pm 0.088$ |
| DenseNet-121 | $0.952 \pm 0.022$ | $\mathbf{0.991 \pm 0.004}$ | $\mathbf{0.960 \pm 0.009}$ | $0.944 \pm 0.048$ | $0.953 \pm 0.020$ | $0.902 \pm 0.041$ |
| Customized | $0.927 \pm 0.026$ | $0.978 \pm 0.012$ | $0.905 \pm 0.074$ | $0.951 \pm 0.031$ | $0.928 \pm 0.041$ | $0.884 \pm 0.002$ |

**Notes.**
Bold text indicate the performance measures of the best-performing model/s.

cells. We also evaluated the performance of pre-trained CNNs by extracting features from different layers in the process of identifying the optimal layer for feature extraction from the underlying data. The naming conventions for these layers are based on the models obtained from Keras® DL library. Layers that gave the best values for the performance metrics are listed in Table 3. Table 4 shows the results obtained by extracting the features from the optimal layers toward classifying the parasitized and uninfected cells.

While performing statistical analyses, we observed that the results of Shapiro–Wilk test were statistically significant for all the performance metrics ($p < 0.05$) to signify that the normality of data has been violated. For this reason, we opted to use the non-parametric Kruskal-Wallis H test. The consolidated results of Kruskal-Wallis H and post-hoc analyses are given in Table 5. We observed that, in terms of accuracy, there existed a statistically significant difference in performance between the different CNNs ($\chi^2(5) = 15.508$, $p = 0.008$). Post-hoc tests further revealed that the statistically significant difference existed between the pre-trained Xception, VGG-16, ResNet-50, and customized

**Table 5  Consolidated results of Kruskal–Wallis H and post-hoc tests.**

| Metric | Kruskal-Wallis H summary | Mean ranks | | Post-hoc |
|---|---|---|---|---|
| Accuracy | $\chi^2(5) = 15.508, p = 0.008$ | AlexNet | 11.20 | Xception & ResNet-50 ($p = 0.005$) |
| | | VGG-16 | 22.30 | Xception & VGG-16 ($p = 0.007$) |
| | | **ResNet-50** | **23.00** | Customized & ResNet-50 ($p = 0.017$) |
| | | Xception | 7.20 | |
| | | DenseNet-121 | 19.60 | |
| | | Customized | 9.70 | |
| AUC | $\chi^2(5) = 18.958, p = 0.002$ | AlexNet | 13.00 | Xception & ResNet-50 ($p = 0.034$) |
| | | VGG-16 | 21.70 | Xception & VGG-16 ($p = 0.030$) |
| | | ResNet-50 | 21.50 | Xception & DenseNet-121 ($p = 0.014$) |
| | | Xception | 4.50 | |
| | | **DenseNet-121** | **22.90** | |
| | | Customized | 9.40 | |
| Sensitivity | $\chi^2(5) = 5.518, p = 0.356$ | AlexNet | 16.20 | – |
| | | VGG-16 | 17.30 | |
| | | ResNet-50 | 15.80 | |
| | | Xception | 11.40 | |
| | | **DenseNet-121** | **21.80** | |
| | | Customized | 10.50 | |
| Specificity | $\chi^2(5) = 6.639, p = 0.249$ | AlexNet | 9.80 | – |
| | | VGG-16 | 20.70 | |
| | | **ResNet-50** | **21.30** | |
| | | Xception | 13.30 | |
| | | DenseNet-121 | 14.10 | |
| | | Customized | 13.80 | |
| F1-score | $\chi^2(5) = 14.798, p = 0.011$ | AlexNet | 11.70 | Xception & ResNet-50 ($p = 0.005$) |
| | | VGG-16 | 22.20 | Xception & VGG-16 ($p = 0.006$) |
| | | **ResNet-50** | **22.60** | Xception & DenseNet-121 ($p = 0.023$) |
| | | Xception | 6.90 | |
| | | DenseNet-121 | 19.50 | |
| | | Customized | 10.10 | |
| MCC | $\chi^2(5) = 14.487, p = 0.013$ | AlexNet | 11.30 | Xception & ResNet-50 ($p = 0.007$) |
| | | VGG-16 | 22.30 | Xception & VGG-16 ($p = 0.008$) |
| | | **ResNet-50** | **22.60** | Xception & DenseNet-121 ($p = 0.034$) |
| | | Xception | 7.60 | Customized & ResNet-50 ($p = 0.021$) |
| | | DenseNet-121 | 19.40 | |
| | | Customized | 9.80 | |

**Notes.**
Bold text indicate the performance measures of the best-performing model/s.

model. In terms of AUC, a statistically significant differenfce was observed ($\chi^2(5) = 18.958$, $p = 0.002$) in the performance between Xception, ResNet-50, VGG-16, and DenseNet-121. Similar results were observed for the F1-score ($\chi^2(5) = 14.798$, $p = 0.011$) and MCC ($\chi^2(5) = 14.487$, $p = 0.013$). No statistically significant difference was observed across the models in terms of sensitivity ($\chi^2(5) = 5.518$, $p = 0.356$) and specificity ($\chi^2(5) = 6.639$,

$p = 0.249$). However, ResNet-50 obtained the highest mean ranks for accuracy, specificity, F1-score, and MCC.

## DISCUSSIONS AND CONCLUSION

The customized model converged to an optimal solution due to hyper-parameter optimization, implicit regularization imposed by smaller convolutional filter sizes and aggressive dropouts in the fully connected layers. Usage of L2 regularization reduced the effect of model overfitting and converging to a better solution (*Simonyan & Zisserman, 2015*).

Each layer of the CNNs produces an activation for the given image. Earlier layers capture primitive features like blobs, edges, and colors that are abstracted by the deeper layers to form higher level features to present a more affluent image representation (*Zeiler & Fergus, 2014*). Studies from the literature reveal that while using pre-trained CNNs for feature extraction, the features are extracted from the layer right before the classification layer (*Razavian et al., 2014*). For this reason, we extracted the features from the second fully connected layer for AlexNet and VGG-16 and the last layer before the final classification layer from Xception, ResNet-50, and DenseNet-121 models. We observed from the patient-level cross-validation studies (Table 2) that ResNet-50 outperformed the customized and other pre-trained CNNs in all performance metrics toward the task of classifying parasitized and uninfected cells. Literature studies reveal that DenseNet-121 outperformed ResNets and other pre-trained CNNs in the ImageNet data classification task (*Huang et al., 2016*). In our case, for the binary task of classifying parasitized and uninfected cells, the variability in data is several orders of magnitude smaller. The top layers of deep CNNs like DenseNet-121 are probably too specialized, progressively more complex and not the best candidate to re-use for the task of our interest. For this reason, we evaluated the performance of pre-trained CNNs by extracting features from different layers in the process of identifying the optimal layer for feature extraction from the underlying data (Table 3). We observed that for the pre-trained CNNs, the performance of the layer before the classification layer was degraded compared to the other layers. In contrast to the results shown in Table 2, DenseNet-121 achieved the best values for sensitivity but demonstrated similar AUC values as ResNet-50 and VGG-16 (Table 4). Both VGG-16 and ResNet-50 were equally accurate and demonstrated equal values for AUC and F1-score. However, ResNet-50 was highly specific, demonstrated high MCC and performed relatively better than the other models under study. These results demonstrate that the final layer of pre-trained CNNs is not always optimal for extracting the features from the underlying data. In our study, features from shallow layers performed better than deep features to aid in improved classification of parasitized and uninfected cells. Literature studies reveal that MCC is an informative single score to evaluate the performance of a binary classifier in a confusion matrix context (*Chicco, 2017*). In this regard, ResNet-50 demonstrated statistically significant MCC metrics as compared to the other models. The consolidated results demonstrated that the pre-trained ResNet-50 relatively outperformed the other models under study toward classifying the parasitized and uninfected cells. While performing Kruskal-Wallis H and post-hoc analyses, we observed that the pre-trained ResNet-50 obtained the highest

**Table 6** Comparison with the state-of-the-art literature.

| Method | Accuracy | Sensitivity | Specificity | AUC | F1-score | MCC |
|---|---|---|---|---|---|---|
| Proposed model (cell level) | **0.986** | **0.981** | **0.992** | **0.999** | **0.987** | **0.972** |
| Proposed model (patient level) | 0.959 | 0.947 | 0.972 | 0.991 | 0.959 | 0.917 |
| *Gopakumar et al. (2018)* | 0.977 | 0.971 | 0.985 | – | – | 0.731 |
| *Bibin, Nair & Punitha (2017)* | 0.963 | 0.976 | 0.959 | – | | |
| *Dong et al. (2017)* | 0.981 | – | – | – | | |
| *Liang et al. (2017)* | 0.973 | 0.969 | 0.977 | – | | |
| *Das et al. (2013)* | 0.840 | **0.981** | 0.689 | – | | |
| *Ross et al. (2006)* | 0.730 | 0.850 | – | – | | |

Notes.
Bold text indicate the performance measures of the best-performing model/s.

mean ranks for accuracy, specificity, F1-score, and MCC. If we were to select a model based on a balance between precision and sensitivity as demonstrated by the F1-score, we could observe that the pre-trained ResNet-50 outperformed the other models under study. We have demonstrated the performance of the models in terms of mean ($\mu$) and standard deviation ($\sigma$) to present a measure of the dispersion in the performance metrics. The pre-trained ResNet-50 outperformed the other models by achieving $0.947 \pm 0.015$ sensitivity and $0.972 \pm 0.10$ specificity. Statistical analyses show that the predictive model could capture all observations within three standard deviations from the mean $[-3\sigma\,3\sigma]$, i.e., the model could exhibit sensitivity and specificity in the range $[0.902\ 0.992]$ and $[0.942\ 1.00]$ respectively. However, our study is focused on disease screening, therefore, the sensitivity metric carries significance. We also determined the number of RBCs that need to be analyzed by the proposed model to confidentially return a positive test result. We used the epiR tools for the Analysis of Epidemiological Data (*Stevenson et al., 2015*) for these computations. The number of cells needed to diagnose (NND) is defined as the number of RBCs to be tested to yield a correct positive test. Youden's index gives a measure of the performance of the model, the value ranges from $-1$ to $+1$ with values closer to 1 for higher values of sensitivity and specificity. With a confidence level (CI) of 0.95 ($p < 0.05$), we found that 11 RBCs need to be tested to return 10 positive results.

To our knowledge, we could find no comparable literature that performed cross-validation studies at the patient level, with a large-scale clinical dataset for the underlying task. For this reason, we also performed cross-validation studies at the cell level and compared with the state-of-the-art (Table 6). In the process, we found that the pre-trained ResNet-50 outperformed the state-of-the-art in all performance metrics. *Das et al. (2013)* achieved similar values for sensitivity with a small-scale dataset but demonstrated suboptimal specificity. The lack of performance at the patient level is attributed to the staining variations between patients. We observed that it is harder for the classifier to learn the different stains, which indicates that we may need to acquire more images with different staining colors for training or apply color normalization techniques. However, by validating the predictive models at the patient-level, which we believe simulate real-world conditions,

we ensure getting rid of bias, reduce overfitting and generalization errors toward optimal model deployment.

We are currently performing pilot studies in deploying the customized and pre-trained DL models into mobile devices and analyzing the performances. From the literature studies, we observed that we could either train/predict on the mobile device or train the model offline and import it to the mobile device for predictions (*Howard et al., 2017*). Currently, Android and IOS ML libraries (like CoreMLStudio) offer the flexibility for dynamic allocation of CPU and GPU based on the computational cost, thus, memory allocation is not an issue. From our pilot studies, we observed that the proposed model occupied only 96 MB and took little RAM to do prediction on the test data. The deployed model could serve as a triage tool and minimize delays in disease-endemic/resource-constrained settings.

### Funding
This work was supported in part by the Intramural Research Program of the National Library of Medicine (NLM), National Institutes of Health (NIH) and the Lister Hill National Center for Biomedical Communications (LHNCBC). The NIH dictated study design, data collection/analysis, decision to publish and/or preparation of the manuscript. The Mahidol-Oxford Tropical Medicine Research Unit is funded by the Wellcome Trust of Great Britain. The Wellcome Trust of Great Britain had no role in study design, data collection and analysis, decision to publish, or preparation of the manuscript.

### Grant Disclosures
The following grant information was disclosed by the authors:
Intramural Research Program of the National Library of Medicine (NLM).
National Institutes of Health (NIH).
Lister Hill National Center for Biomedical Communications (LHNCBC).
Wellcome Trust of Great Britain.

### Competing Interests
The authors declare there are no competing interests.

### Author Contributions
- Sivaramakrishnan Rajaraman conceived and designed the experiments, performed the experiments, analyzed the data, prepared figures and/or tables, authored or reviewed drafts of the paper, approved the final draft.
- Sameer K. Antani and Stefan Jaeger conceived and designed the experiments, authored or reviewed drafts of the paper, approved the final draft.
- Mahdieh Poostchi performed the experiments, prepared figures and/or tables, authored or reviewed drafts of the paper, approved the final draft.

- Kamolrat Silamut, Md. A. Hossain and Richard J. Maude analyzed the data, contributed reagents/materials/analysis tools, authored or reviewed drafts of the paper, approved the final draft.
- George R. Thoma conceived and designed the experiments, analyzed the data, authored or reviewed drafts of the paper, approved the final draft.

## Ethics

The following information was supplied relating to ethical approvals (i.e., approving body and any reference numbers):

The Institutional Review Board (IRB) at the National Library of Medicine (NLM), National Institutes of Health (NIH), granted approval to carry out the study within its facilities.

## Data Availability

The segmented cells from the thin blood smear slide images for the parasitized and uninfected classes are made available at https://ceb.nlm.nih.gov/repositories/malaria-datasets/. The dataset contains a total of 27,558 cell images with equal instances of parasitized and uninfected cells.

## Supplemental Information

Supplemental information for this article can be found online at http://dx.doi.org/10.7717/peerj.4568#supplemental-information.

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
