# Peer review of "Pre-trained convolutional neural networks as feature extractors toward improved malaria parasite detection in thin blood smear images"

_PeerJ, doi:10.7717/peerj.4568_

## Round 0.1 · original submission · Major Revisions

The reviewers and I find the manuscript interesting, but in need of major revisions before it can be accepted. In addition to the in-depth constructive feedback by the reviewers, I would like to add the following.

The strength of the paper is not in the novelty of the methods, but the validation on a larger clinical dataset. Remove any novelty claims and focus on the clinical validation aspect.

The validation, as one of the reviewers mentioned, is not clear and could be biased. I want to see a very clear description of the train and test data and how exactly the performance was derived. I want to see cross-validation on a patient basis.

The paper is about classification with various methods. Yet I see only one ROC curve. I expect to see more, one curve per method. Furthermore, I expect to see a proper statistical analysis of ROC outcome. Not Kruskal Wallis, but proper ROC analysis tools, consult a statistician.

The order of the manuscript is not OK. The statistical analysis section comes after the results section which is very unusual. The results describe statistical tests. Please read the ICMJE recommendations about how to write a medical image analysis paper.

·

Basic reporting

I am very happy to see that the work done by the authors are tested with sufficient dataset collected from real patients unlike many manuscripts (which often report results on cultured cell dataset / an insufficient dataset). The manuscript is written in good English.

Experimental design

Yes. The the work done by authors meet the aim and scope of the journal. Though the novelty is minimal (other than they have worked on real patient dataset), I think the manuscript can be substantially improved by addressing the comments that I have raised in 'validity of the findings' section reported below.

Validity of the findings

1. As you have pointed out (Lines 104 - 106), Liang et al. has reported that they have found (at least for the dataset that they were using), the custom designed CNN worked much better than the pretrained model (here, your choice). It would be interesting to see the behavior for your dataset. i.e., you design a CNN that works on the same set of images used to test the pretrained model and report the accuracy (sensitivity, specificity).

2. As you have pointed out in 141-142, RBC segmentation is an important and challenging task. You are claiming that the segmentation algorithm is novel. But this reviewer could not assess the novelty (from your part). For me, it appears that you have just combined two popular models. This thought, may be due to little details that you have provided on cell segmentation. Please elaborate.

For example, in Fig.1 you can see neutrophils. How good your algorithm is in segmenting WBCs? What strategy you have used to distinguish a WBC (say a lymphocyte) from a heavily infected RBC?

You must give details on cell segmentation and on the technique that you have used to assess the accuracy of segmentation? For example, manual annotation with the Dice coefficient as performance measure will do an independent evaluation.

3. Have you come across the following recent papers.?

"Image analysis and machine learning for detecting malaria, 2017 Elsevier Translational Research"

"Convolutional neural network-based malaria diagnosis from focus stack of blood smear images acquired using custom-built slide scanner. 2017 Wiley journal of Biophotonics"

4. Line 160 - 163, the authors are claiming that they have ensured sufficient data for training (generalized training) and sufficient data for testing? How did you conclude that the particular split is sufficient?

5. As you have pointed out in Line 107, you are also doing spatial scaling. Did you study its effect? It would be interesting to see the result of this study.

6. Can you justify/at least speculate on the behavior that you have reported in Lines 213 & 214?

7. When it comes to practical testing scenario, the number of infected cells is going to be very less compared to the uninfected cells (say in 1% parasitemia). Since the number of RBCs is very high in 1 microlitre of blood (~5 million), we need a system offering ~1 specificity and a very high sensitivity (say 0.995). In such case (especially for a medical system) rather than simply reporting sensitivity and specificity it is preferred to report a combined measure such as MCC as well.

8.In the practical scenario (in Point # 7), it would be interesting and beneficial if you add the following analysis

Given a system (with p sensitivity, r specificity), how many RBCs need to be analyzed to confidently decide that a person is infected by malaria (given he has q% parasitemia).

With the above reasoning/study, report how many RBCs you need to analyze to confidently decide (statistically significant) a person (with q% parasitemea) is infected or not by the best system that you have developed (say with DenseNet 121).

Also report by the same system, what is the lower level of parasitemea you can diagnose.

Additional comments

I appreciate your work. As I have pointed out in above sections, the novelty is limited. However if you address all of my comments listed above, it is going to be a good paper and I believe that you can contribute great to the field (in building a usable device for malaria screening).

Reviewer 2 ·

Basic reporting

This article evaluated the performance of five well known pre-trained CNN models as feature extractors for automated malaria screening. This paper is well written and the performance of the proposed method seems to be good.

Experimental design

In general, the experimental design was good and clearly written. However, this article needs some major revisions.

Validity of the findings

No Comment

Additional comments

Major points:
1) In the article, you have mentioned that:
"In this study, we instantiated the convolutional part of the pre-trained CNNs, everything up to the fully-connected layers and trained a fully-connected model on top of the stored features." (lines 180-182)

However, after reviewing your code (pre_trained_CNN_feature_extraction.py), I don't see the parts by which you store features, or freezing (fixing) the pre-trained layers in order to train just your defined fully-connected model. What I see in your code is that you have retrained (not fine-tune) the pre-trained model alongside your randomly initialized fully-connected model.

When you adding a randomly initialized fully-connected model on top of a pre-trained model and training them all together (without freezing pre-trained layers), the large gradient updates triggered by the randomly initialized weights would wreck the learned weights in the pre-trained model. And as a result, you cannot claim that you have used a pre-trained model.

2) It worries me that you might have used your test data as your validation data too. In line 161, you have mentioned that:
"In total, we had 24,760 cell images for training and 2730 for testing."
and in your code (load_data.py - lines 9, 10 and 11) the numbers are:
nb_train_samples = 24760 # training samples
nb_valid_samples = 2730 # validation samples
nb_test_samples = 2730 # test samples
and there is no validation folder in your shared dataset file. Could you explain how you chose validation images?

Minor points:
3) Can you provide examples of images on which the algorithm has failed?
4) In the first row of Figure 1, the input image (A) does not match with other images (B, C and D)

---

## Round 0.2 · Minor Revisions

The rebuttal addressed many of the comments of the reviewers, with a few valid remarks left to answer. Please consider these. Regarding the performance on a case basis, I think it will suffice to state what the performance of human readers is on classifying cells. As stated in the introduction, your method should be better.

The conclusion section needs adaptation. The paper states that it revalidates previously published methods using cross-validated, patient-based analysis, which is commendable. The valid hypothesis being that this might show that predicted performance decrease. In your study, you show a decrease in performance, but you did not compare to train/test set based analysis like in the previous literature. Comparing to previous studies and attributing the performance decrease is weak, as dataset and methods are always somewhat different. I am interested to see how you are going to solve this issue.

The last section in the results section (336-343) should move to the discussion section. It is a bit weird to discuss your results in before drawing any conclusions.

I like the fact that your method is intended to run on a smartphone. This probably will lead to using smaller models that have worse results. It wasn't clear when reading the text if this is the case in your study when comparing to other studies. This could also be part of the discussion section.

·

Basic reporting

The manuscript is improved considerably.

Experimental design

Experimental design looks fine now, except for the validation of the segmentation/cell detection accuracy.

Validity of the findings

The findings are valid but need to be more specific in reporting especially for the qn 8 in their rebuttal letter.

Additional comments

For qn. 1 in rebuttal letter: Did you consider any other model? Why did you choose the specific architecture?)

For qn. 2 : From the points, I believe that you did not evaluate the accuracy of segmentation. May be because you don't have manually annotated images for segmentation. I wonder how you measured the precision and recall for cell detection just by identifying the centroids. How did you compare the true centroids with the detected one? i.e., did you compare the exact pixel location or you had a tolerance? Make these things explicit.

For qn. 3: Since there are many experiments done in the paper, when you report sensitivity and specificity, you are bound to specify the number of samples in the test set and for what cross validation/experiment the reported result is.

For qn 8: I am not completely satisfied with the response for Qn 8 in the rebuttal. Suppose you are analyzing 1000 cells from 1% parasitaemic individual. Out of these cells, it is reasonable to believe that 990 cells are healthy and 10 cells are infected. Your best system with 0.947 sensitivity and 0.972 specificity will then behave something like this : It will detect 1 out of 10 infected cells as healthy and detect 27 healthy cells as infected. This means that for all samples, the system will identify infected cells. This is true even for the samples from a perfectly healthy individual. So the question is, how will it be useful? Or how many cells has to be analysed to ensure that 99.99% chance is there that we have identified really an infected cell among the so many identified infected cells.

Every run in cross validation experiment (provided in Table 1) ideally should take same number of samples? Why there is a difference in number of samples used across the runs? How did you select the samples for each run?

Any reported result in this paper to substantiate the claim made in line 291?

How does the diagnostic odds ratio (line 298) differ from sensitivity?


Minor comments:
Fig. 1 : 2nd row Images placed one over the other and need to be corrected.
Fig.2 looks a little odd. It could have drawn more professionally.

Please proof read the document carefully. There are missing spaces (such as line 41), repeated words (such as improving improved in line 184)

Reviewer 2 ·

Basic reporting

No Comment

Experimental design

No Comment

Validity of the findings

No Comment

Additional comments

I think the paper has now been substantially improved and deserves publication.

---

## Round 0.3 · accepted · Accept

All comments were addressed appropriately.

Reviewer 2 ·

Basic reporting

No Comment

Experimental design

No Comment

Validity of the findings

No Comment